# PIXEL CHEM: A REPRESENTATION FOR PREDICTING MATERIAL PROPERTIES WITH NEURAL NETWORK

## ABSTRACT

In this work, we developed a new representation of the chemical information for the machine learning models, with benefits from both the real space (R-space) and energy space (K-space). Different from the previous symmetric matrix presentations, the charge transfer channel based on Pauling's electronegativity is derived from the dependence on real space distance and orbitals for the heteroatomic structures. This representation can work for the bulk materials as well as the low dimensional nanomaterials, and can map the R-space and K-space into the pixel space (P-space) by training and testing 130k structures. P-space can well reproduce the R-space quantities within error 0.53. This new asymmetric matrix representation doubles the information storage than the previous symmetric representations. This work provides a new dimension for the computational chemistry towards the machine learning architecture.

## 1 INTRODUCTION

The most common way of representing material structures is using the coordinates of atoms together with the periodic unit cell vectors and categorized this information as unstructured data (i.e. `cif` files)(Lam Pham et al., 2017). However, those unstructured data cannot put into neural network directly and need to be reformatted as matrices for the advanced GPU based neural network computational architecture, which is extremely robust in pixel processing.(**?**) As Scholkopf & Smola (2002) indicated, when using Kernel-based machine learning models, in order to reach a fast and accurate prediction of first principle properties, a single Hilbert space of the atomistic system is required, in which regression is carried out. In all above situation, an ideal representation needs to possess several essential characters like invariant, unique, continuous, general and efficient etc.(Rupp, 2015; Bartok et al., 2013; von Lilienfeld et al., 2015): In addition, apart from the R-space information, other K-space information might be more important for the overall property of material, especially when the research target relates to the band gap or excitation.

Various kinds of representation have been developed and reported. The Coulomb matrix representation (CM, Rupp (2015); Faber et al. (2015); Rupp et al. (2012)) coded nuclear charge and atom displacement inside to predict atomization energies and formation energies. Based on it, bag of bond (BOB)(Hansen et al., 2015) representation constructed an effective pairwise interatomic potential to strip down pair-wise variant of CM, which improved the predictive model remarkably. Many-body tensor representation (MBTR, Huo & Rupp (2017)) is derived from 2 above and reached a significant drop in error rate. Symmetry functions(Behler, 2011; Eshet et al., 2010; 2012) is another method which focuses on the representation of local chemical environment of atoms and maps this representation to atomic energy. Recently, based on Simplified molecular-input line-entry system (SMILES, Weininger (1988)) representation, a new 3N-MCTS module based on Mote-Carlo tree search is proposed by Segler et al. (2018). This module keeps only the structure-connection information and trained large dataset to provide supplementary instructions for retrosynthetic analysis. In addition, representations like bonding-angular machine learning(Hansen et al., 2015) and others(Huang & von Lilienfeld, 2016; Schutt et al., 2014) have also been reported and tested. However, most of these representations only satisfy those requirements partially, in which unique and continuous principles are often violated.

The most severe shortcoming for both CM and BoB is that they focus on predicting the extentive quantity,in R space like structural energy but left other intensive quantities in K space like band gap.

Though some reported researches also included other properties(Huang & von Lilienfeld, 2016), their error rates are way larger comparing with energy. Overall, the R-space and K-space are still separated for this kind of representation. The possible improvements could be made by endorsing more R/K space information to the targeted representations. Recall that CM only coded atomic number together with distance, BoB added roughly designed bonds energy in compliment, both failed to represent most of the properties in K-space. Therefore, defining a new representation with enough physical identity from R/K space is a vital task for material informatics to ensure the uniqueness and continuity.

## 2 DEFINITION

### 2.1 PIXEL CHEM REPRESENTATION FOR FINITE STRUCTURES

Here in this work, we proposed representations with projected both R and K space chemical information into the pixel space (Pixel Chem). We start from the property of different atoms, mainly binding strength and charge transfer ability. To describe these, we first introduce two new matrices, $C$ and $B$. $C$, which mainly reflects the charge transfer ability, is based on electron affinity energy and ionization energy of atoms. Inspired by Mullikan Electronegativity(Mulliken, 1934), it is defined as $C_{i,j} = (IE_{i,:} + EA_{:,j})/2$. As for binding strength, $B_{i,j}(U_{\text{Bond}})$ is imported, which represents the typical bond energy formed by two atoms. Note that though $B_{i,j} = B_{j,i}$, $C_{i,j} \neq C_{j,i}$, which means this matrix is not symmetric between diagonal. Finally, we proposed a displacement matrix $D_{i,j} = D_{j,i} = \exp(-|r_i - r_j|/a)$ to represent the relative position of different atoms, where $|r_i - r_j|$ represent the Euclidean distance between two atoms and $a$ is the interaction strength coefficient trained by machine learning methods. To mix all the things up, we proposed a charge & energy matrix $E_{i,j}$ which can be produced by multiplying $D_{i,j}$ with $C_{i,j}^T$ and $B_{i,j}^T$ as follows:

$$E = D \times C^T \times B^T \tag{1}$$

This operation ensures that for every element $E'$ in $E_{i,j}$, there will be a product of the $D'$ in $D_{i,j}$ and $B'$ in $B_{i,j}^T$ corresponding to the same atom. Therefore, for all the atoms, we have:

$$E'_{i,j} = D'_{i,j} \cdot C'_{i,j} \cdot B'_{i,j} \tag{2}$$

Atomic orbits information, including arrangement, position and number, is another important factor that highly influence the property of a structure. In order to represent those information, we first construct a $1 \times 10$ sized orbital charge vector $o_i = [N_{\sum core}, N_s, N_{px}, N_{py} \ldots N_{dyz}, N_{dxz}]$ (core stands for all the core electrons) is created to represent (mainly) the electron configuration of every atom in valance electron orbitals. The binding energy $B_E$ of each orbital also needs to be considered to distinguish those orbitals in different chemical environments. We therefore product the $\exp(-B_E^{\text{orbit}})$ of every orbital in $o_i$ to get an energy encoded orbital matrix $o_{\text{BE}i}$, where:

$$o_{\text{BE}i} = [N_{\sum core} \cdot \exp(-B_E^{\sum core}), N_s \cdot \exp(-B_E^s), \cdots N_{dyz} \cdot \exp(-B_E^{d_{x^2-y^2}})] \tag{3}$$

To better represent the chemical environment of each atom, especially the different condition of each orbital, we hereby introduce two closely related methods, tight-binding method and linear combination of atomic orbitals (LCAO, Slater & Koster (1954)) method, to our representation model. With the existence of atomic Hamiltonian, this matrix can precisely describe the interaction between two atoms orbital by orbital. In detail, a $10 \times 10$ sized angular interaction matrix $A_{i,j}$, representing from all core electrons to $d_z$ orbital electrons, is implemented for all atoms in one specific structure, which formats as:

$$\begin{bmatrix} Int(i[\sum core], j[\sum core]) & Int(i[\sum core], b[s]) & \cdots & Int(i[\sum core], j[d_{x^2-y^2}]) \\ Int(i[s], j[\sum core]) & Int(i[s], j[s]) & \cdots & Int(i[s], j[d_{x^2-y^2}]) \\ \vdots & \vdots & \ddots & \vdots \\ Int(i[d_{x^2-y^2}], j[\sum core]) & Int(i[d_{x^2-y^2}], j[s]) & \cdots & Int(i[d_{x^2-y^2}], j[d_{x^2-y^2}]) \end{bmatrix}_{10 \times 10} \tag{4}$$

which $i[m]$ represent the $m$ orbital of $i$ atom.

Recalled 3, every element inside the matrix of atoms, shown as $Int(A[\text{orbit}a], B[\text{orbit}b])$, represent the interaction strength between two orbitals which belong to two atoms respectively, it is yielded out as follows:

$$Int(i[m], j[n]) = N_{i[m]} \cdot \exp(-B_E^{i[m]}) \cdot N_{j[n]} \cdot \exp(-B_E^{j[n]}) \cdot Coeff \tag{5}$$

Which can be summarized as:

$$Int(i[m], j[n]) = \boldsymbol{o}_{\text{BE}i}[m] \cdot \boldsymbol{o}_{\text{BE}j}[n] \cdot Coeff \tag{6}$$

The coefficient inside this formula is defined by the position of two atoms, the direction of two target orbits and the energy of them. To be specific, the calculation was primarily based on the Slater-Koster Approximation(Hehre et al., 1972), which was designed to solve the overlapping area between two selected orbits. Some orbits, such as $p_x$, $p_y$ and $p_z$ orbits for the same atom, are perpendicular to each other, which will result in $0$ at the corresponding element. The calculation detail could be found in supplementary notes. In all, this angular interaction matrix expanded the previous matrix for 10 times, which encodes more physical meanings, especially the Hamiltonian of each atom, which could benefit the learning efficiency and accuracy of our training model.

To combine all the angular matrices $\boldsymbol{A}_{i,j}$ with the previous $\boldsymbol{E}$. We product every element $\boldsymbol{E}_{i,j}$ with its corresponding $\boldsymbol{A}_{i,j}$ to get new element $\boldsymbol{M}_{i,j}$. Note that new element $\boldsymbol{M}_{i,j}$ is also a $10 \times 10$ matrix. Therefore, $\boldsymbol{M}$ will be the final Pixel Chem representation, denoted as:

$$\boldsymbol{M}_{i,j} = \boldsymbol{E}_{i,j} \cdot \boldsymbol{A}_{i,j} \tag{7}$$

Which can also be written as:

$$\boldsymbol{M}_{i,j} = \boldsymbol{D}_{i,j} \cdot \boldsymbol{C}_{i,j} \cdot \boldsymbol{B}_{i,j} \cdot \boldsymbol{A}_{i,j} \tag{8}$$

Moreover, when implementing Pixel Chem into organic structures, this representation could be simplified because most of the elements in organic structures are from the first 3 periods which means they do not have $d$ orbitals. This simplification results in a smaller matrix size and could handle the representation of organic molecules in a faster way. When considering organic structures, hybridization, especially for carbon and nitrogen, the type of chemical bonds need to be considered. We hereby implement an adjoint classifier to estimate the chemical state for each carbon, which will return the bond type and hybridization state of carbon atoms. Take sp3 as an example, instead of which proposed above, the $\boldsymbol{A}_{i,j}$ for carbon atom will be transferred to:

$$\begin{bmatrix} Int(a[\sum \text{core}], b[\sum \text{core}]) & Int(i[\sum \text{core}], j[s]) & \cdots & Int(i[\sum \text{core}], j[pz]) \\ Int(i[s], j[\sum \text{core}]) & Int(i[s], j[s]) & \cdots & Int(i[s], j[pz]) \\ \vdots & \vdots & \ddots & \vdots \\ Int(i[pz], j[\sum \text{core}]) & Int(i[pz], j[s]) & \cdots & Int(i[pz], j[pz]) \end{bmatrix}_{5 \times 5} \tag{9}$$

Hereby the $\boldsymbol{o}_{\text{BE}i} = [N_{\sum \text{core}} \cdot \exp(-B_E^{\sum \text{core}}), N_s \cdot \exp(-B_E^s), \cdots, N_{pz} \cdot \exp(-B_E^{pz})]$.

as 4 sp3 orbitals present the same property. $B_E^{\text{sp3}}$ will also take the place of their initial values. Figure 1 shows different orbital matrix for different hybridization state.

To illustrate this representation in a straightforward way, we visualized the pixel chem using `matplotlib` in Figure 2. Each colored square represents a numerical value. Black color stands for absolute 0, which also means there is no electron in the corresponding orbital. Some of the off-diagonal and diagonal parts are carried out with the explanation, which are used to represent atoms properties (for diagonal parts) or interaction strength between atoms (for off-diagonal parts).

## 2.2 FROM FINITE TO PERIODIC STRUCTURES

Apart from finite structures like fullerene, the bulk periodic structures like diamond and quasi 2D layered periodic structures such as graphite or multi-layer graphene could also be represented by Pixel Chem representation, as shown in Figure 3 (a) to (c). In this situation, we calculate the interaction including atoms belonging to itself and two adjacent primitive cells, $-z$ and $+z$ respectively. For those adjacent cells, $\boldsymbol{D}_{i,j}$ is the only difference and it was altered as $\boldsymbol{D}_{i,j} = \boldsymbol{D}_{j,i} =$

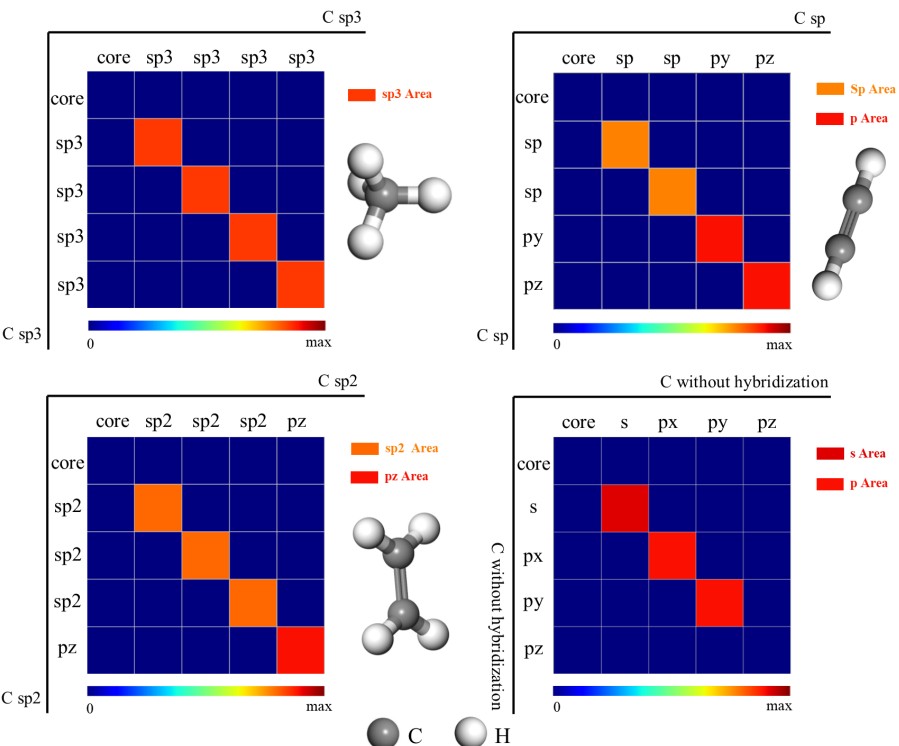

Figure 1: Visualized orbital matrix representation for different hybridization states of carbon. We use a color spectrum from blue to red to represent the value of the different elements. These kinds of matrices lie on the diagonal parts of Pixel Chem. On off-diagonal parts, elements distance matrix is smaller, so the color will be bluer in most of the cases. All the s and p orbits, including sp/sp2/sp3 hybridization orbits, have no overlap area between each other, which yields out all 0 values in off-diagonal parts of four matrices.

$\sum_{k=-1}^{1} \exp(-|\boldsymbol{r}_i - \boldsymbol{r}_j| + kc)$ ($c$ is the lattice parameter, $k = \{-1, 0, 1\}$ for adjacent cells) to include all interaction parts. Note that the $\boldsymbol{D}_{i,j}$ in diagonal part will also be larger, which simply indicates whether this matrix stands for a periodic structure or a finite structure. By measuring this part, this structure's lattice constant, $C$, could be determined. Moreover, this representation comes together with an adjacent cell interaction strength that could be measured directly from the matrix. Generally, this methodology also works for periodic structures which have 0 to 2-dimensional periodicity. In those cases, the adjacent cells in the corresponding dimension will be calculated in the same way. Figure 3 lists 4 other different structures for a better view of the unity of Pixel Chem. Among those 4 structures, all except $C_{60}$ are periodic and diamond has periodicity for all three dimensions. This can be directly seen from the diagonal parts where the corresponding orbital interaction value of periodic dimensions will increase. Note that for the bottom two isoelectronic structures, apart from the orbital difference, graphite is strictly symmetric across diagonal while BN is asymmetric because of the different charge transfer ability.

## 2.3 BENEFITS

Comparing with other representations in previous works, Pixel Chem has enormous benefits. Firstly, it satisfied all the requirements indicated by Rupp (2015); Bartok et al. (2013); von Lilienfeld et al. (2015), which are (1) invariant, (2) unique, (3) continuous, (4) general, (5) fast and efficient for calculation. For the invariant property, the neural network we designed is order-uncorrelated. Meanwhile, the variety of position and atom property of different structures could secure the uniqueness even for isomers. For (3), though the representation is discrete, the interaction between every two atoms and their major orbitals are considered with no information gap. For (4), this representation

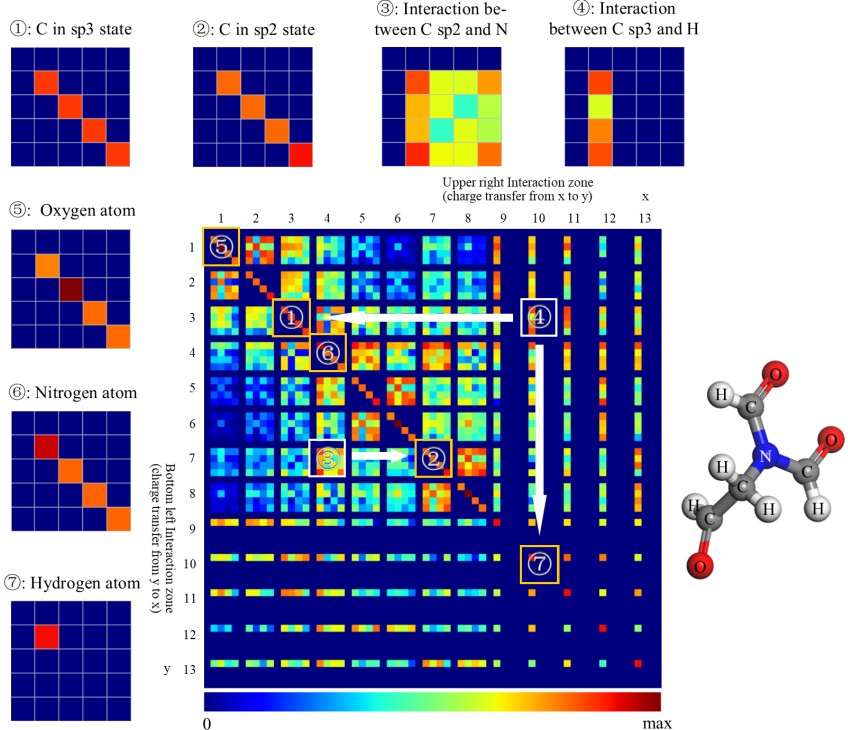

Figure 2: Visualized Pixel Chem for a typical molecule in QM9 dataset, $C_4H_5O_3N$. Note that upper right and lower left parts are not perfectly symmetric because the charge transfer ability $C_{Ei,j}$ differs. The color of different pixels is together influenced by orbitals, bonds, angles and charges.

could handle both finite and periodic structures with only a little bit of difference in definition. On the contrary, the CM and most of the previous modules could only handle finite structures. For (5), compared with conventional methods, which typically needs more than 10 minutes for a structure with 20 atoms, our neural network could output the predicted value in less than 1 second if the model has been well trained. Moreover, the matrix representation in the Pixel Chem is asymmetrically mirrored by the diagonals due to the charge transfer directions in heteroatomic structures, doubled of the information storage size compared with the previous matrix representation. In conclusion, comparing with CM, though Pixel Chem has similar shape and arrangement, the information contains inside is more sufficient as both orbital information, orbital angle, and charge transferability are encoded.

## 3 NETWORK

What we designed is a Pixel Chemistry network (PCnet) which can learn a representation for the prediction of molecular properties such as energy and band gap. The network reflects basic physical laws in molecule including invariance to atom indexing. All the prediction for different properties is rotationally invariant.

### 3.1 ARCHITECTURE

The Figure 4 shows an overview of the architecture of PCnet, which is inspired by Deep Tensor Neural Network (DTNN, Schütt et al. (2017)). Interactions between atoms are modeled by three interaction blocks. The final prediction of the properties is obtained after atom-wise updates of the feature representation and pooling of the resulting atom-wise energy. We will discuss the different components of the network in the following.

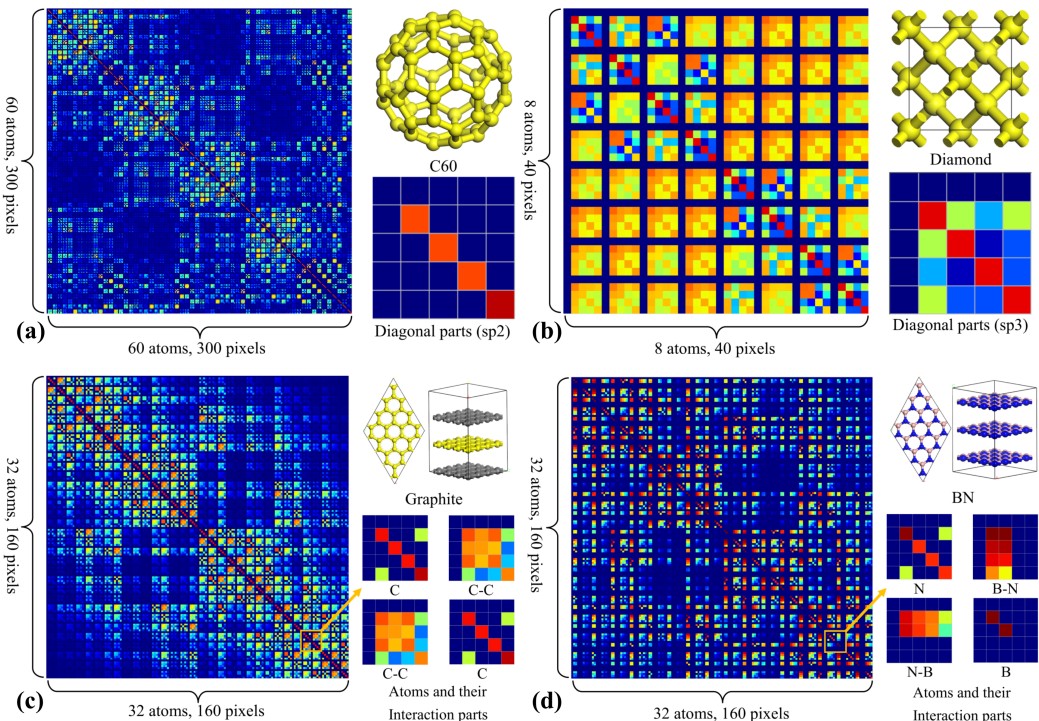

Figure 3: (a) (b) (c) (d): 3 carbon allotropes and a boron nitride (BN) structure generated by Pixel Chem. (a) $C_{60}$, 60 Carbon atoms without periodicity, symmetric, (b) Diamond, 8 Carbon atoms with periodicity on 3 dimensions, symmetric, (c) Graphite, 32 Carbon atoms with $z$ direction periodicity, symmetric, (d) BN, 16 Boron atoms and 16 Nitrogen atoms with $z$ direction periodicity, a graphite isoelectronic, asymmetric.

**Molecular representation** A molecule in a certain conformation can be described uniquely by a set of $n$ atoms with data of atoms $\boldsymbol{A} = (\boldsymbol{A}_1, \ldots, \boldsymbol{A}_n)$ and interaction between each two of all atoms. We divided the interaction into upper part $\boldsymbol{U} = (\boldsymbol{U}_1, \boldsymbol{U}_2, \ldots, \boldsymbol{U}_n)$ and lower part $\boldsymbol{L} = (\boldsymbol{L}_1, \boldsymbol{L}_2, \ldots, \boldsymbol{L}_n)$ by the position in the matrix. $\boldsymbol{U}_i = (\boldsymbol{U}_{i,1}, \cdots, \boldsymbol{U}_{i,(i-1)}, \boldsymbol{U}_{i,(i+1)}, \cdots, \boldsymbol{U}_{i,n})$ represent all the upper part interaction between atom $i$ and other atoms, $\boldsymbol{L}_i = (\boldsymbol{L}_{i,1}, \cdots, \boldsymbol{L}_{i,(i-1)}, \boldsymbol{L}_{i,(i+1)}, \cdots, \boldsymbol{L}_{i,n})$ represent all the lower part interaction between atom $i$ and other atoms.

Through the layers of PCnet, we use a tuple of features $\boldsymbol{X}^l = (\boldsymbol{x}_1^l, \ldots, \boldsymbol{x}_n^l)$ to represent the atoms or interaction data, where $\boldsymbol{x}_i^l \in \mathbb{R}^F$ with the number of atoms $n$, the current layer $l$, the number of features map $F$.

**Atom-wise layer** A dense layer that is applied to the feature $\boldsymbol{x}_i^l$ of atom $i$ separately with learnable weight $\boldsymbol{W}^l$ and learnable bias $\boldsymbol{b}^l$ in layer $l$:

$$\boldsymbol{x}_i^{l+1} = \boldsymbol{W}^l \boldsymbol{x}_i^l + \boldsymbol{b}^l$$

The atom-wise layer is designed to represent the combination of the origin feature map. PCnet can work for molecules with any size since weights of atom-wise layers are shared across atoms.

**Interaction** The interaction block is designed to update the representations of atoms based on the interaction with other atoms. The interaction block uses a residual connection inspired by ResNet(He et al., 2016):

$$\boldsymbol{x}_i^{l+1} = \boldsymbol{x}_i^l + \boldsymbol{r}_{i1}^l + \boldsymbol{r}_{i2}^l$$

As shown in the interaction block in figure 4, the residual $\boldsymbol{r}_{i1}^l$ and $\boldsymbol{r}_{i2}^l$ are computed through two dense layers and a `tanh` non-linearity with lower and upper part interaction as input.

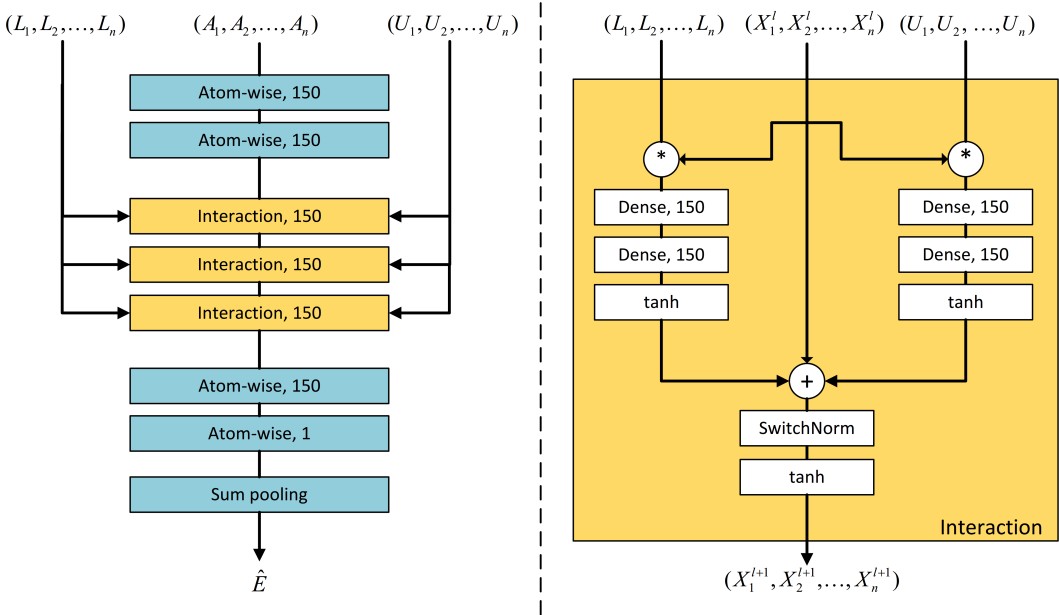

Figure 4: Illustration of PCnet with an architectural overview(left) and the interaction block(right).

After the residual, we use switchable normalization layer (SwitchNorm, Luo et al. (2018)) to normalize the feature $x_i^l$:

$$x_i^{l+1} = \gamma \frac{x_i^l - \mu}{\sqrt{\sigma^2 + \epsilon}} + \beta$$

where $\gamma$ and $\beta$ are a learnable scale and a learnable shift parameter respectively. $\epsilon$ is a small constant to prevent numerical instability. The SwitchNorm is continuing with a $\tanh$ non-linearity.

We keep the number of feature map to be constant $F = 150$ throughout the whole network except the last atom-wise layer. All atom-wise layers and dense layers are followed by a learnable PReLU non-linearity(Xu et al., 2015). The last atom-wise layer is followed by a hyperbolic tangent non-linearity. The feature output from the last atom-wise layer represents the differences between the ground state energy and the energy contributed in the molecule, followed by a sum pooling layer which sum all atoms' contributed energy to get the predicted energy.

## 3.2 TRAINING

We apply PCnet to the quantum chemistry dataset QM9(Ramakrishnan et al., 2014). QM9 has 133885 molecules. We remove 3423 unreasonable molecules with largest absolute cohesive energy, which may have N-N-N bonds in a single molecule. The cohesive energy is defined as the energy of molecule subtracts to the sum of atoms' energies in the molecule. Cohesive energy is useful infiltrating molecules by their stability. This energy is usually negative, and because of the subtraction operation, it can show the part of the energy used to combine atoms together. A molecule is considered more stable if less energy is used to combine atoms together.

We randomly chose 10000 molecules for validation, 10000 molecules for testing, and used remains for training. For energy prediction, the model was trained using SGD with the ADAM optimizer(Kingma & Ba, 2014) with 32 molecules per mini-batch for 1 million steps. We used $10^{-3}$ as the initial learning rate and an exponential learning rate decay with rate 0.96 every 2 epochs (220924 steps). All targets were normalized to mean 0 and variance 1. We minimized the mean absolute error between predicted energy $\hat{E}$ and true energy $E$. The same method is also applied to predict other properties.

Table 1: Comparison of mean absolute error of previous approaches (left) and our network (right) in $\mathrm{kCal/mol}$.(Gilmer et al., 2017)

| Target | BAML | BOB | CM | ECFP4 | HDAD | GC | GG-NN | DTNN | enn-s2s | PCnet |
|--------|------|-----|-----|-------|------|-----|-------|------|---------|-------|
| U0 | 1.21 | 1.43 | 2.98 | 85.01 | 0.58 | 3.02 | 0.83 | 0.84[1] | 0.45 | 0.53 |
| gap | 3.28 | 3.41 | 5.32 | 3.86 | 2.49 | 1.78 | 1.70 | N/A | 1.60 | 2.13 |
| HOMO | 2.20 | 2.20 | 3.09 | 2.89 | 1.54 | 1.18 | 1.17 | N/A | 0.99 | 1.44 |
| LUMO | 2.76 | 2.74 | 4.26 | 3.10 | 1.96 | 1.10 | 1.08 | N/A | 0.87 | 1.26 |

### 3.3 RESULT

In Table 1 we compare the performance of PCnet and the previous approaches. These baselines include 5 different hand engineered molecular representations, which then get fed through a standard, off-the-shelf classifier. These input representations include the Coulomb Matrix (CM, Neese (2003)), BoB, Bonds Angles, Machine Learning (BAML, Huang & von Lilienfeld (2016)), Extended Connectivity Fingerprints (ECPF4, Rogers & Hahn (2010)), and "Projected Histograms" (HDAD, Faber et al. (2017a)) representations. In addition to these hand engineered features we include the Molecular Graph Convolutions model (GC, Kearnes et al. (2016)), the original GG-NN model(Li et al., 2015) trained with distance bins, DTNN and enn-s2s from Message Passing Neural Networks (MPNNs, Gilmer et al. (2017)).

We can see from Table 1 that PCnet can achieve a low error bars (0.53) in the R-space U0 energy quantity due to the directional coulomb interaction in the representation without any additional load for imposing more geometric information than HDAD. The K-space accuracy is already below the DFT (B3LYP, Faber et al. (2017b)) level errors. But for other properties, the errors of PCnet prediction results are higher than some method, and PCnet needs further improvement.

## 4 CONCLUSION

We proposed a new representation of material structures, named as Pixel Chem and designed a PCnet neural network to process this module. We also introduced a new pixel space apart from R-space and K-space in conventional chemistry representations. Distinct from those representations which focus mainly on numerical approach, our module covers both finite and periodic systems. In the future, this representation has a large prospect. First, the combination of parameters can still be optimized to carry more information with less space. Moreover, more physical and chemical properties, such as Young's modulus and melting point, can be predicted using the same way. Finally, it can be wildly implemented in other fields. One example is computational biology, where it can be used to construct larger material structures.

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
