# OpenReview forum: "Pixel Chem: A Representation for Predicting Material Properties with Neural Network"
_ICLR.cc/2019/Conference_

### Official Review · AnonReviewer2 · 2018-10-24
**Poorly motivated representation with unsubstantiated claims**

**Rating:** 3
**Confidence:** 5

**Review:**

This paper proposes a neural network architecture PCnet for the prediction of intensive and extensive chemical properties of molecules and materials. The authors claim that the use of prior chemical knowledge such as Mulliken electronegativity, bond strength and orbital information improves prediction accuracy. While the idea of incorporating chemical domain knowledge in the interactions of an atomistic neural network is interesting in principle, this paper has severe issues ranging from presentation over the proposed approach to the results.

First and foremost, I would like to point out that the results in Table 1 are cherry-picked since the authors fail to cite neural network architectures that outperform their approach, e.g. for U0: 0.45 kcal/mol [MPNN, Gilmer et al 2017, ICML], 0.31 kcal/mol [SchNet, Schütt et al, NIPS 30, 2017], 0.26 kcal/mol [HIP-NN, Lubbers et al., JCP 148, 2018]. This is especially apparent since Figure 4 is obviously inspired by Fig. 1 in [SchNet, Schütt et al, JCP 148, 2018].

The authors use a variety of heuristics and approximations such as a "charge transfer ability", bond strength, exponential decay of distances and overlaps of atomic orbitals which are multiplied "to mix all things up", to arrive at the PixelChem representation which is then fed into an atomistic neural network (PCnet). Combining these chemical features in such a way is neither well-motivated, nor does it lead to an improvement in accuracy compared to state-of-the-art networks.
Even for the intensive properties (gap, HOMO, LUMO), where PCnet is supposed to have an advantage due to its use of orbital information, MPNN, SchNet and even GC and GG-NN in Table 1 outperform the proposed approach. Parameterization of the chemical features and training the PCnet end-to-end might have improved results and seems like a missed opportunity.

Further issues:
- The manuscript is riddled with typos, grammatical errors as well as confusing sentences.
- The authors claim that PCnet is applicable to periodic structures, however, this is never demonstrated. Beyond that their definition of periodic PixelChem does only include adjacent cells, while for a unique representation more cells might be required.
- The "benefits" listed in Section 2.3 compare selectively to previous work. E.g., invariances, uniqueness, asymmetric interactions are also fullfiled by the neural networks listed above. A comparison of the PixelChem representation to the Coulomb matrix is not sufficient here.
- The PCnet architecture uses PReLU nonlinearities. While this is fine for equilibrium predictions, for other configurations this prohibits the prediction of a smooth PES.

Overall, I believe that it is important to incorporate chemical knowledge into neural networks. However, neither the approach nor the results convince me that this has been achieved here.

---

> ### Author Response · Authors · 2018-11-19
> **We will improve our work based on your constructive review**
>
> Thanks for your constructive review.
> We have added the result of [1] in our comparison, and fixed typos in our paper.
> And we notice that our PCnet underperformance some neural network currently, and we are trying to improve the performance based on your review.
> Our Pixel Chem is applicable to periodic structures like the demonstration in Figure 3. Due to the dataset of periodic molecules are building, we have not included more information about periodic structures in the paper currently.
> Thank you again for the useful review, and we will continuing perfect our work according to your comments. We always welcome more constructive comments about our idea.
>
>         Best Regards,
> Authors of paper 372
>
> [1] Gilmer, Justin, et al. "Neural message passing for quantum chemistry." arXiv preprint arXiv:1704.01212 (2017).

---

### Official Review · AnonReviewer1 · 2018-11-07
**Large overlap with prior work which is not cited, bordering on plagiarism. Architecture not interesting, and results significantly underperform prior work.**

**Rating:** 1
**Confidence:** 5

**Review:**

This paper proposes an architecture for learning representations on molecules. The paper contains a number of typos, and presents results and an entire paragraph that is a near duplicate copy from prior work (which is not cited). In particular Table 1 is a near copy of Table 2 appearing in [1].

I find it particularly suspicious that the authors have a near copy of this prior table, which reports the ratio of MAE to chemical accuracy, and not MAE directly. The authors have the exact same numbers as this prior Table 2 but claim they are reporting MAE directly. Despite this table being a direct copy, it conveniently omits the columns from [1] which outperform the results presented here.

Also the paragraph describing this table is nearly identical to the paragraph in [1], that is the authors write
"These baselines include 5 different hand engineered molecular representations, which then get fed through a standard, off-the-shelf classifier. These input representations include the Coulomb Matrix (CM, Neese (2003)), BoB, Bonds Angles, Machine Learning (BAML, Huang & von Lilienfeld (2016)), Extended Connectivity Fingerprints (ECPF4, Rogers & Hahn (2010)), and Projected Histograms (HDAD, Faber et al. (2017a)) representations. In addition to these hand engineered features we include the Molecular Graph Convolutions model (GC, Kearnes et al. (2016)), the original GG-NN model(Li et al., 2015) trained with distance bins and DTNN."

In [1] it was written

"These baselines include 5 different hand engineered molecular representations, which then get fed through a standard, off-the-shelf classifier. These input representations include the Coulomb Matrix (CM, Rupp et al. (2012)), Bag of Bonds (BoB, Hansen et al. (2015)), Bonds Angles, Machine Learning (BAML, Huang & von Lilienfeld (2016)), Extended Connectivity Fingerprints (ECPF4, Rogers & Hahn (2010)), and “Projected Histograms” (HDAD, Faber et al. (2017)) representations. In addition to these hand engineered features we include two existing baseline MPNNs, the Molecular Graph Convolutions model (GC) from Kearnes et al. (2016), and the original GG-NN model Li et al. (2016) trained with distance bins."

The footnote in the table is also a duplicate copy of the footnote in [1].

Theirs: "As reported in DTNN. The model was trained on a different train/test split with 100k training samples vs
about 110k used in our experiments."

[1]: "As reported in Schutt et al. ¨ (2017). The model was trained on a different train/test split with 100k training samples vs 110k used in our experiments."


The proposed method itself is a variant of the MPNN framework introduced in [1], yet underperforms the original results in [1], as well as improved MPNNs (or GNNs) shown in [2,3].

1. https://arxiv.org/pdf/1704.01212.pdf
2. http://papers.nips.cc/paper/6700-schnet-a-continuous-filter-convolutional-neural-network-for-modeling-quantum-interactions
3. https://arxiv.org/pdf/1806.03146.pdf

---

> ### Author Response · Authors · 2018-11-19
> **We have added miss citations and reference**
>
> Thanks for your review.
> We are sorry to miss citations and reference of [1], and now we add it into our paper.
> We cite the data in Table 2 of [1] in our paper.  The data in Table 2 of [1]  is calculated by using chemical accuracy shown in Table 1 in supplementary of [1]. We use it to calculate MAE from the ratio of MAE to chemical accuracy and write in Table 1  in our paper. The calculated MAEs have the same number with the ratio of MAE to chemical accuracy, which may confuse you.
> We welcome some constructive ideas about our work further.
>
>         Best Regards,
> Authors of paper 372
>
> [1] Gilmer, Justin, et al. "Neural message passing for quantum chemistry." arXiv preprint arXiv:1704.01212 (2017).

---

### Official Review · AnonReviewer3 · 2018-11-09
**Review of "Pixel Chem: A Representation for Predicting Material Properties with Neural Network"**

**Rating:** 3
**Confidence:** 3

**Review:**

This paper suggests a new architecture for representing and predicting properties of molecules ("Pixel Chem"). The authors report that this architecture produces better results than previous methods on one of the QM9 dataset properties (U0) but performs worse than many of the other reported methods on other quantities.

Overall, this paper is extremely unclear, contains typos in most sentences, and provides insufficient justification for the both a) the design choices made and b) the claims made wrt the performance of the model. Additionally, at least one critical baseline [1] is missing which outperforms the proposed model. In order to improve this paper, I would suggest the authors do the following:

1) Heavily edit and rewrite the paper focusing on clarity of communication.
2) Provide justifications for why design choices were made. For example, the authors state that they include a charge and energy matrix to "mix all the things up." Why does it make sense to combine these factors in this way?
3) Provide a more comprehensive evaluation of the model, showing improved performance across more than one target and including appropriate baselines, including [1].

[1] Gilmer, Justin, et al. "Neural message passing for quantum chemistry." arXiv preprint arXiv:1704.01212 (2017).

---

> ### Author Response · Authors · 2018-11-19
> **We are working to improve our work**
>
> Thanks for your review.
> According to your words, we had fix typos in our paper and add MPNN as a baseline. We realize that our PCnet underperformance some neural network currently, and we are trying to improve the performance based on your review.
> We always welcome constructive ideas about our work.
>
>     Best Regards,
> Authors of paper 372

---

### Public Comment · ~Longyang_Yao1 · 2018-10-25
**How to select molecules to remove?**

A good idea to represent molecules. I am also working in QM9 database. I notice that you choose 3423 molecules to remove is selected by cohesive energy. The description of this deletion is too short. Could you provide more information about it?

---

> ### Author Response · Authors · 2018-10-26
> **Spin is better than cohesive energy in considering deletion**
>
> Thank you for your attention to our paper. Actually, we found that considering cohesive energy is not the best way to delete inappropriate data from QM9.  It is more reasonable to delete the structures with spin. The spin comes from some single electrons, which means those nucleus does not satisfy the 8 outer electrons rule and will result in a property of instability. According to the path we creating matrices, theoretically, the PCnet can have better adaption to structures after deleting data with spin and actually we have an improvement of about 2% in prediction accuracy compared to the method in the current version. We will update the related content during the rebuttal period, and you can keep track of our newest version. Hope you can give us more feedback in the future.
>
> 	Best Regards,
> Authors of paper 372

---

### Public Comment · ~Qi_Liu1 · 2018-10-25
**other prospects?**

Very interesting work. My research focuses on computational material science and I have read many papers focusing on this area, like DTNN. I noticed that the main part in this work is you used a new method to represent organic structures and picked up some physical quantities. Could you explain more about why you choose those quantities rather than others? And also, will it be able to predict more complex properties in the future, like Raman spectrum (I know it will be difficult though) ?

---

> ### Author Response · Authors · 2018-10-26
> **This can be done with more properties encoded and some revise with our presentation**
>
> Thanks for your comments.
> For the first question, as we stated in the paper, our main focus is on implementing detailed orbital information and charge transfer ability into the previous representation method. Therefore, we chose bonds energy, ionization energy as well as electron affinity to reinforce the charge transfer ability, and take distance together with orbital direction for orbital information.
> For the second question, we did think about it at the very beginning. As Ramon spectrum mainly observe vibration and rotation modes of a structure, we first need to find or construct a database with enough data, whether theoretically or experimentally, which is difficult. Second, we need to find a way to encode these messages into our representation, which may require more tests. In all, it is possible to include other properties as you stated and more work is needed to be done in the future.
> Hope you can give us more feedback in the future.
>
> 	Best Regards,
> Authors of paper 372

---

### Meta-Review · Area_Chair1 · 2018-12-14
**Serious concerns, should be rejected**

**Confidence:** 5
**Recommendation:** Reject

**Metareview:**

I would like to highlight to the PCs that reviewers highlighted clear evidence of plagiarism from prior work, which I was able to easily verify (a full paragraph of text was copied, word-for-word, from a paper describing one of the baselines the current work compares against). Further, all reviewers unanimously agreed that the paper was poorly written, and contains no useful advances for the ICLR audience. I recommend a rejection, and further, examination by the PCs of the conduct of the authors.